# Evaluation of Xpert MTB/RIF Assay, MTB Culture and Line Probe Assay for the Detection of MDR Tuberculosis in AFB Smear Negative Specimens

**DOI:** 10.3390/diseases10040082

**Published:** 2022-10-06

**Authors:** Chandri Lama, Sanjib Adhikari, Sanjeep Sapkota, Ramesh Sharma Regmi, Gokarna Raj Ghimire, Megha Raj Banjara, Prakash Ghimire, Komal Raj Rijal

**Affiliations:** 1Central Department of Microbiology, Tribhuvan University, Kirtipur 44618, Nepal; 2Guangzhou Institute of Biomedicine and Health, Chinese Academy of Sciences, Guangzhou 510530, China; 3National TB Reference Laboratory, National Tuberculosis Centre, Thimi 44600, Nepal

**Keywords:** *Mycobacterium tuberculosis*, smear-negative specimens, Xpert MTB/RIF, LPA, sensitivity, specificity

## Abstract

The global burden of tuberculosis (TB), particularly with multidrug resistance (MDR), is escalating and has become a major health challenge. It is well known that acid-fast bacilli (AFB) smear-negative TB patients are the major source of spreading TB to healthy individuals when left untreated. Early diagnosis of TB and rapid detection of drug resistance are important for the proper management of drug-resistant TB (DR-TB). Therefore, a laboratory based cross-sectional study was conducted from July to December 2019 at the National Tuberculosis Centre, Thimi, Nepal, with the objective of evaluating the diagnostic performance of Xpert MTB/RIF assay, *Mycobacterium tuberculosis* (MTB) culture and line probe assay (LPA) for the detection of MDR-TB in AFB smear-negative sputum samples. We evaluated a total of 222 AFB smear-negative sputum specimens, of which 21.6% (n = 48) showed MTB positive with Xpert MTB/RIF assay and, while culturing on Lowenstein–Jensen (LJ) media, 21.2% (n = 47) were MTB culture positive. The sensitivity, specificity, PPV and NPV at 95% confidence interval of Xpert MTB/RIF assay on diagnosing *M. tuberculosis* from smear-negative specimens were 73% (57–84), 92% (87–96), 71% (59–81) and 93% (89–95), respectively. In addition, the sensitivity of Xpert MTB/RIF assay and LPA in detecting rifampicin resistance was 75% (42–94, 95% CI) and 91.67% (62–99, 95% CI), respectively. The current study also assessed a significant association between the occurrence of pulmonary tuberculosis with different age group, TB history and alcohol consumption. These findings indicate that Xpert MTB/RIF assay and LPA are appropriate methods for early detection and accurate diagnosis of TB and RIF mono-resistant cases.

## 1. Introduction

Tuberculosis (TB) is a serious public health problem caused by *Mycobacterium tuberculosis* (MTB) and a leading cause of mortality from a single infection since 2014. According to WHO, an estimated 10.0 million people fell ill with TB and 1.4 million died of it in 2019 [1]. In Nepal, TB ranks among the top 10 leading causes of death with 32,043 newly diagnosed TB cases reported in the fiscal year 2018/19 [2].

The burden of DR TB is not as high as the regional or global burden in Nepal. There are estimated around 1500 (0.84–2.4) cases of DR TB annually. However, 350 to 450 MDR TB cases are notified annually. In 2019, 392 MDR cases were enrolled (out of 635 notified), a modest decrease from the previous year. The lack of availability and early screening of suspects with rapid DST may still be the main reasons for this stagnation of DR TB cases in Nepal [2]. WHO’s End TB Strategy aims to end TB epidemic by 2035 [3]; however, relying mostly on the old microscopic diagnostic testing with low sensitivity, a century-old less effective vaccine, Bacille Calmette–Guerin, and old drugs are the facts that hamper global TB control programs [4].

The emergence of MDR-TB defined as TB caused by strains resistant to at least two first-line drugs, INH and RIF; and extensively drug-resistant tuberculosis (XDR-TB) defined as TB caused by strains resistant to the two above-mentioned drugs; to at least one fluoroquinolone; and to at least one of three injectable second line drugs (amikacin, kanamycin and capreomycin) makes for a worrisome situation [5]. Since the WHO assembly has declared MDR-TB as the global health emergency, the mainstay for its control is the rapid and accurate identification of infected individuals.

Sputum smear microscopy through Ziehl–Neelsen (ZN) staining is extensively used in developing countries for routine TB diagnosis due to cost effectiveness and high specificity, and does not require sophisticated equipment [6]. Despite that effectiveness, smear microscopy is less sensitive, compared to other diagnostic tools. Smear-negative pulmonary TB accounts for 30–60% of all pulmonary TB cases [7]. The gold standard approach for TB diagnosis is a culture technique utilizing Lowenstein–Jensen (LJ) medium for mycobacterial growth [8]. It takes a longer period, generally 3–4 weeks, and has a high sensitivity, but it requires a sophisticated laboratory [9]. This might result in the delayed initiation of necessary treatment, which raises the potential of dissemination and the development of drug resistance, owing to the initiation of inappropriate medication.

As almost 13% of TB transmission occurs with smear-negative, culture-positive TB patients [8], the risk of disease transmission by AFB-negative cases to healthy individuals should not be overlooked. Despite tremendous efforts, the incidence of TB in developing countries, such as Nepal, is still a major public health concern and, therefore, requires use of highly sensitive and specific techniques for the early detection of MTB. Xpert MTB/RIF assay and LPA are reliable molecular-based methods that are capable of detecting resistance against rifampicin. There are very few pieces of research completed so far in Nepal that have attempted to evaluate the diagnostic efficacy of these assays, taking into account the smear-negative cases. In light of this, the current study aimed at evaluating the diagnostic performance of Xpert MTB/RIF assay, MTB culture and LPA in detecting MTB in smear-negative PTB cases.

## 2. Materials and Methods

### 2.1. Study Period, Design and Sample Size

A cross-sectional laboratory-based study was conducted at National Tuberculosis Centre (NTC) at Thimi, Bhaktapur, Nepal, from July to December 2019. NTC is specific TB clinic in Nepal. All the patients showing the symptoms of TB visiting NTC during the study period were recruited in the study. Altogether 1895 patients attended the center during the study period. Out of them, 1673 showed smear positive results, and thus they were excluded from the study. Remaining 222 patients’ sample gave smear negative results and thus they were included for the study purpose.

### 2.2. Specimen and Data Collection

Sputum and bronco-alveolar lavage (BAL) specimens were collected from the patients with signs and symptoms suggestive of PTB or with a chest X-ray showing abnormalities suggestive of TB. Only AFB smear-negative specimens were processed for the identification and evaluation of performance characteristics of the Xpert MTB/RIF and LPA. About 3–5 mL of early morning sputum sample was collected in wide mouthed, transparent, plastic, sterile leakage-proof and screw-capped 50 mL falcon tube. The tube was then labelled with the patients’ names, dates and serial numbers. The clinical specimens accepting Bartlett criteria were included in the study [10]. Sputum mixed with saliva, pus, blood and specimen less than 3–5 mL were excluded. Specimens were rejected if they contained excessive oropharyngeal contaminants. A pre-structured questionnaire was used to collect data on demographics, as well as different risk factors, such as smoking and alcohol consuming habits.

### 2.3. Sample Processing

The N-acetyl-l-cysteine-sodium citrate-NaOH (NALC-NaOH) technique was used to disinfect all the samples initially [11]. After centrifugation at 3000× *g* for 15 min, the samples were decanted, and the pellets were re-suspended in 3 mL of phosphate-buffered solution. Several aliquots from each sample were made and used for the florescent microscopy, culture, Xpert and LPA. ZN-staining for smear microscopy was performed following the WHO recommended protocol [12]. The smear-negative sputum sample was mixed with phosphate-buffered saline (PBS), vortexed and left for 15 min at room temperature, and a cartridge containing this mixture was placed in the Xpert machine. The interpretation of data from Xpert MTB/RIF tests is software based and takes around 90 min to process the sample [13]. Another aliquot of the decontaminated samples was then cultured on LJ media. The inoculated slants were incubated at 37 °C. After a week, the caps of the tubes were tightened securely and further incubated in an upright position at 37 °C for 8 weeks. Colony morphology, ZN staining and MPT64 antigen tests using the reference strain of *M. tuberculosis* H_37_Rv were done for the confirmation of *M. tuberculosis* [14].

### 2.4. Drug Susceptibility Testing (DST)

Drug susceptibility testing was done by three methods: Conventional DST, Xpert MTB/RIF and line probe assay (LPA). Conventional DST was performed in the LJ medium following the protocol mentioned in a previous study [15]. LPA testing was performed in separate rooms with restricted access and unidirectional workflow [16]. DNA extraction was performed in the BSL3 laboratory. Master mix preparation, PCR and hybridization assays were performed in designated rooms. LPA was carried out with the extracted DNA using GenoType^®^ MTBDR*plus* (Hain Lifescience GmbH, Nehren, Germany) for detection of rifampicin resistance according to the manufacturer’s instructions.

### 2.5. Quality Control

Laboratory equipment, such as incubator, autoclave, hot air oven and refrigerator, were regularly monitored for their efficiency. Reagents and media were meticulously checked for the expiry date and proper storage conditions. They were properly labelled with the preparation date. Control strain of *M. tuberculosis* H_37_R_v_ was used for the confirmation of *M. tuberculosis* colonies on LJ medium.

### 2.6. Data Analysis

Data were analyzed by using SPSS software for windows (v25). The chi-square test (*χ*^2^) was estimated and a value of *p* ≤ 0.05 was considered significant wherever applicable. Sensitivity, specificity, positive predictive value, negative predictive value and accuracy were calculated to determine the performance of the test.

## 3. Results

A total of 1895 samples were collected from pulmonary tuberculosis suspected patients (1055 male and 840 female patients) during the study period. Out of the total samples collected, all the smear-negative samples (222 in number) were purposively taken as the study population, which comprised of 23 (10.4%) BAL samples and 199 (89.6%) sputum samples, respectively. The remaining 1673 patients’ sample showed smear positive result, and were, hence, excluded in the study. Among 222 fluorescent microscopy negative samples subjected to further investigations, 47 samples were culture positive on LJ media and 48 samples were positive on Xpert MTB/RIF assay (Table 1).

### 3.1. Distribution of Tuberculosis Patients with Various Attributes

Out of 222 study participants enrolled in the study, 35 (22.88%) males were Xpert MTB/RIF assay-positive, whereas the highest percentage of culture positive samples (23.19%) were reported from samples taken from females (*p* > 0.05). A higher prevalence of the MTB infection was reported from the specimen collected from the age group 15–54 in Xpert MTB/RIF assay (27.69%) and culture (26.92%) (*p* < 0.05). Patients with a previous TB history have a high rate of TB infection, as detected by Xpert MTB/RIF assay (60.71%) and culture (64.29%) (*p* < 0.05). Smokers were more likely to be infected with TB than non-smokers (*p* > 0.05). Moreover, the highest percentage of the alcohol consumers was noted to have TB infection, as reported by Xpert MTB/RIF assay (59.52%) and culture (57.14%) (*p* < 0.05) (Table 2).

### 3.2. Performance of Xpert MTB/RIF Assay, as Compared to Gold Standard Culture Method

The performance characteristics of the Xpert MTB/RIF assay are shown in Table 3. Out of 222 smear-negative specimens, 47 were culture positive, while 48 specimens were positive by Xpert MTB/RIF assay. Among 47 culture positive specimens, 13 specimens were Xpert MTB/RIF negative. The sensitivity, specificity, PPV and NPV at 95% confidence interval of Xpert MTB/RIF assay were 73% (57–84), 92% (87–96), 71% (59–81) and 93% (89–95), respectively. The overall accuracy of this test was 88%.

### 3.3. Summary of Anti-Tubercular Drug Susceptibility Test

A total of 47 culture positive cases were subjected to drug susceptibility tests, of which 36 (76.60%) were sensitive to rifampicin and 11 (23.40%) showed mono-resistance pattern, i.e., rifampicin resistance by Xpert MTB/RIF assay. On convectional drug testing, 35 (74.47%) were sensitive to rifampicin, whereas 12 (25.53%) were rifampicin resistant. Among 47 MTB culture positive specimens subjected to LPA assay, 35 (72.34%) specimens were RIF sensitive and 13 (27.66%) cases were RIF resistance in LPA drug susceptibility testing method.

### 3.4. Performance of Xpert MTB/RIF Assay and LPA in Detecting Rifampicin Resistance

Diagnostic efficacy of Xpert MTB/RIF assay and LPA in detecting rifampicin resistance was assessed in 47 culture positive specimens, taking conventional drug susceptibility test as the gold standard. Sensitivity, specificity, PPV and NPV of Xpert MTB/RIF assay at 95% confidence interval were 75% (42–94), 94.29% (80–99), 81.82% (52–94) and 91.67% (80–96), respectively. Likewise, sensitivity, specificity, PPV and NPV of LPA at 95% confidence interval were 91.67% (62–99), 97.14% (85–99), 91.67% (61–98) and 97.14% (83–99), respectively (Table 4).

## 4. Discussion

With the objective of evaluating the diagnostic performance of Xpert MTB/RIF assay, MTB culture and LPA for detection of MDR-TB, we investigated 222 smear-negative sputum samples obtained from 1895 pulmonary tuberculosis suspected patients. Out of 222 smear-negative specimens examined by fluorescence microscopy, 48 (21.6%) were confirmed as TB positive by Xpert MTB/RIF assay, whereas 47 (21.2%) were confirmed as TB positive by MTB culture on LJ medium. A similar study done in Spain reported MTB culture positivity in 108 (64.3%) smear-negative specimens of which 82 specimens were Xpert MTB/RIF Ultra assay positive [17]. In addition, a retrospective study revealed 85 (69.7%) *Mycobacterium tuberculosis* complex (MTC) culture positive samples in a total of 125 smear-negative clinical samples [18]. The discovery of the Xpert MTB/RIF assay is vital, especially in developing countries, as the assay aids in the rapid prognosis of the smear-negative TB cases, which were beforehand a venture for the TB control programs.

A higher percentage of males were Xpert MTB/RIF assay positive (22.8% male vs. 18.8% female), whereas a higher percentage of culture positive sample was observed with females (23.2% female vs. 20.3% male) (*p* > 0.05). The current study demonstrated considerable difference among gender in the occurrence of TB infection. This might be due to the socio-economic and cultural, as well as immunological, factors of patients contributing to the development of tuberculosis [19,20]. Individuals belonging to the age group 15–54 were found more likely to get infected with tuberculosis than people of other age groups. The result of this study is in line with the previous studies [21,22]. Since people of the productive age group are highly active and their mobility is more frequent, the chances of exposure are also high, which supports the evidence of acquiring the disease. Individuals with a previous TB infection were seen more likely to develop tuberculosis. This may be due to the fact that TB infections can occur, owing to the relapse of the same strain [23]. This study reported a higher rate of TB infections among smokers and alcohol consumers. This finding is in commensurate with the previous studies [24,25,26]. The reason can be attributed to the impaired mucosal secretion, diminishing alveolar macrophage phagocytic ability and reduction in immunological responses in smokers and alcoholics.

In the current study, Xpert MTB/RIF assay showed low sensitivity (73%) and high specificity (92%). In comparison to several other studies, the sensitivity we found was low for Xpert MTB/RIF assay. A study from Pakistan has demonstrated a slightly higher sensitivity, compared to our study [27]. However, a similar study in Ethiopia has even reported a lower sensitivity than our study [28]. The difference in sensitivities might be due to the genetic variation among the study populations. In addition, the assay conditions and technical expertise might have also influenced the results.

Anti-tubercular drug susceptibility testing revealed that 25.53% of the isolates were rifampicin resistant. A previous study done in Kathmandu, Nepal has shown a fairly high rate of rifampicin resistance (81.6%) amongst *Mycobacterium tuberculosis* culture positive isolates [29]. In contrast, a piece of research performed in Pakistan reported rifampicin resistance in 15.4% isolates [30]. In the present study, a relatively lower rate of rifampicin resistance (23.40%) was reported by Xpert MTB/RIF assay than the previous studies [29,31]. The explanation for this variation can be attributed to the mutation in other regions instead of hot spots, which could be solved by sequencing the *rpoB* gene. The sensitivity and specificity of the Xpert MTB/RIF assay, as compared to gold standard conventional drug susceptibility test for detecting rifampicin resistance, were 75% and 94.29%, respectively. A comparable study conducted in Nepal documented a higher sensitivity (98.60%) and specificity (100%) than our study [29]. On the other hand, the sensitivity and specificity of the LPA were 91.67% and 97.14%. The performance characteristics of this assay were, however, higher than the study performed in Ethiopia [32]. Geographical locations of sample collection, variances in sampling technique, MDR-TB and mutations within the *rpoB* gene might be the responsible factors for variations in diagnostic performance characteristics [33]. Both Xpert MTB/RIF and LPA assays are associated with less biological hazards and short turnaround time, in comparison to the conventional DST. These assays can be a very useful tool in controlling MDR-TB since smear-negative cases are responsible for the spread of drug resistance in the community.

## 5. Conclusions

Several smear negative samples, when examined by Xpert MTB/RIF, LPA and culture methods, exhibited positive results, which crucially signifies that we are missing actual TB patients while we use microscopy only. In other words, patients whose samples are AFB negative could still have active tuberculosis. Furthermore, considering MTB culture as the reference assay standard, our study showed that LPA has better diagnostic performance characteristics than the Xpert MTB/RIF assay. In addition, regarding RIF mono-resistance detection, LPA outperformed the Xpert MTB/ RIF assay, and thus it is a better alternative when it comes to detecting RIF resistance.

## Figures and Tables

**Table 1 diseases-10-00082-t001:** Comparison between Fluorescence AFB Sputum Smear Microscopy, Xpert MTB/RIF Assay and MTB Culture.

Results	Methods		
	Fluorescence Microscopy	Xpert MTB/RIF Assay	MTB Culture on LJ Medium
Positive	0	48 (21.62%)	47 (21.17%)
Negative	222 (100%)	174 (78.38%)	166 (74.78%)
Contamination	0	0	9 (4.05%)

**Table 2 diseases-10-00082-t002:** Distribution of tuberculosis infection, with respect to various attributes.

Attributes	Sample Size	Presence of *M. tuberculosis* Detected by			
		Xpert MTB/RIF Assay n (%)	*p*-Value	Culture n (%)	*p*-Value
Gender					
Male	153	35 (22.88)	0.5	31 (20.26)	0.62
Female	69	13 (18.84)		16 (23.19)	
Age group					
≤14	4	1 (25.00)	0.027 *	1 (25.00)	0.037 *
15–54	130	36 (27.69)		35 (26.92)	
≥55	88	11 (12.50)		11 (12.50)	
Previous TB history					
Yes	28	17 (60.71)	0.0001 *	18 (64.29)	0.0001 *
No	194	31 (15.98)		29 (14.95)	
Smoking					
Yes	49	11 (22.45)	0.26	9 (18.37)	0.718
No	173	27 (15.61)		28 (16.18)	
Alcohol consumption					
Yes	42	25 (59.52)	0.0001 *	24 (57.14)	0.0001 *
No	180	23 (12.78)		23 (12.78)	

* Significant at 5% level of significance.

**Table 3 diseases-10-00082-t003:** Comparison of Xpert MTB/RIF with reference to culture.

Method	Culture			*p*-Value	Sensitivity	Specificity	PPV	NPV
Xpert MTB/RIF	Positive	Negative	Total					
Positive	34 (70.8)	14 (29.2)	48	0.01	73%	92%	71%	93%
Negative	13 (7.5)	161 (92.5)	174					

**Table 4 diseases-10-00082-t004:** Performance of Xpert MTB/RIF assay and LPA in detecting rifampicin resistance, as compared to gold standard conventional drug susceptibility test.

Test Performed	Sensitivity % (95% CI)	Specificity % (95% CI)	PPV % (95% CI)	NPV % (95% CI)	Accuracy %
Xpert MTB/RIF assay	75 (42–94)	94.29 (80–99)	81.82(52–94)	91.67 (80–96)	89
LPA	91.67 (62–99)	97.14 (85–99)	91.67 (61–98)	97.14 (83–99)	95

## Data Availability

All the data are available within the manuscript.

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
