# Peer review of "Evaluation of Xpert MTB/RIF Assay, MTB Culture and Line Probe Assay for the Detection of MDR Tuberculosis in AFB Smear Negative Specimens"

_diseases, 2022, doi:10.3390/diseases10040082_

Round 1
Reviewer 1 Report
1 Introduction
In Nepal 420 of the 1500 estimated (M)DR cases (28%) are detected and thus 72% not detected. This is a good statement and the study would gain to show, what proportion of the missed (M)DR cases can be found, when applying Xpert test, i.s.o. smear microscopy on the presumptive TB cases. This is to my opinion, the strength of the study, and the authors should focus on that
There are more global bottlenecks, but it would be more interesting, what the issues are in Nepal, why the 2035 targets would not be met
Pls indicate (including a reference) When did WHO assembly declared MDR-TB as the global health emergency,
The statement: Sputum smear microscopy through Ziehl– Neelsen (ZN) staining is extensively used in developing countries is not true, and is also based on a old reference.
The reference to a WHO report of 2010 is not reflecting the situation of today!!
Although the study objective: In light of this, the current study aimed at evaluating the diagnostic performance of GeneXpert MTB/RIF assay, MTB culture and LPA in detecting MTB in smear-negative PTB cases is correct good, the background information states a few major errors:
· The statement: “It is well known that AFB smear-negative TB patients are the major source of spreading TB to healthy individuals when left untreated” is definitely not true. The infectiousness of AFB smear positive patients is much higher
· Also the statement: “The continuously increasing frequency of TB in developing countries like Nepal…”is not true. According to WHO global report 2021 the estimated burden of TB in Nepal is decreasing since 2000
2.1 How many patient in total were attending the clinic ?
3. Results:
Out of 1895 samples collected, 222 smear-negative samples (from 222 patients?) were taken as the study population. This is 12% Were all remaining 1673 patients smear positive? Not clear how these 222 patients were selected: where these the only samples smear negative?, pls make clear
You may mention the characteristics ( age. And sex distribution) of the 222 study population
Table 1 is not clear: I don’t understand the line positive ( smear positive?) and 0 among fluorescence microscopy, As Xpert and culture results among the 222 smear negative cases are overlapping, it would be helpful toe add a third column with Xpert/Culture positivity
Also helpful would be to mention the definition of “smoking” and “alcohol use” and in the methods how it was ascertained
3.3. when talking about sensitivity PPV etc, please indicate what is used as the Gold Standard?
23 -27% among the 222 were Rifampicin resistance, this is quite high ? But here you
4 Discussion:
I don’t understand the statement: “the current study reported no considerable difference among gender in the occurrence of TB infection; there were about twice as much males compared to females in the study population. Moreover, the study was focusses on test performance among a specific group (smear negatives symptomatic) but not on occurrence of TB infection. You may delete this section.
The study shows that “In the current study, Xpert MTB/RIF assay showed low sensitivity (73%)”..also the authors note that smoking and alcohol use are risk factors: the study may improve if sensitivity is shown by different risk groups including age and gender.
The authors mention that; “The difference in sensitivities might be due to the genetic variation among the study populations. Besides, the assay conditions and technical expertise might have also influenced the results” I think, the differences have more to do, with differences in the characteristic of the study population. Therefor it so important to describe in the introduction, the setting of the clinic ( is it a regular clinic? or a specific TB clinic, where people attend with specific TB symptoms? Also, were the 222 the only one who were smear negative among the 1895 attendees? This is
The statement A previous study done in Kathmandu, Nepal has shown a fairly high rate of Rifampicin resistance (81.6%) amongst Mycobacterium spp. [29]. Pls explain what is spp? 82% is otherwise very high, and actually impossible
5. Conclusion
“Xpert MTB/RIF assay can be a useful diagnostic option for the patients suspected of pulmonary tuberculosis-either smear positive or smear negative-with rapidity and also detects Rifampicin mono-resistance,…..” Can be an option…this statement might be a correct conclusion, but it disregards that since long time Xpert is already the international standard to test any TB suspects (presumptive TB patients), irrespective the smear status. The authors suggests that first a smear test need to be investigated, but Xpert test should replace smear microscopy for the diagnosis of TB.
Try to link the conclusion with the main problems mentioned in the introduction: there are many (M)DR missed but when applying Xpert to all symptomatic patients attending the clinic, this proportion may reduce to …..%
Reviewer 2 Report
Summary: The authors evaluated GeneXpert MTB/RIF Assay, MTB Culture and Line Probe Assay for the Detection of MDR Tuberculosis in AFB Smear Negative Specimen. MDR TB is difficult to treat and stressing to patients and TB programmes. Identifying patients with MDR-TB would support early treatment and subsequently reduce the likelihood of transmission. Therefore, research on the diagnostic performance of the tools that can effectively diagnose MDR-TB is crucial. However, there are areas that authors could improve:
Abstract: The statement, 'current study also assessed a significant association between the occurrence of pulmonary tuberculosis with different age group, TB history and alcohol consumption (p>0.05)’ is not clear as it does not present any results. The p-value of each association and statistical test applied, should be stated.
Introduction: The Napal MDR-TB national data cited by authors is of 2018. This almost 5 years, isn't there a more recent data to cite?
In paragraph four, the statement, 'the gold standard approach for TB diagnosis is a culture technique utilizing Lowenstein–Jensen (LJ) medium for mycobacterial growth [8] cites a 2007 paper which compared performance of BACTEC 460 and LJ medium for isolation of M. tuberculosis. In all aspects BACTEC 460 supported MTB growth 3x more samples than LJ culture. So, the authors have misquoted the paper. Secondly, this is an over 13-year old paper and therefore is not an effective citation in as far as current gold standard for TB diagnosis is concerned.
In paragraph five, no citation is given for the following statements:
1. It is well known that AFB smear-negative TB patients are the major source of spread-ing TB to healthy individuals when left untreated.
2. GeneXpert MTB/RIF assay and LPA are reliable molecular based methods that are capable of detecting resistance against Rifampicin.
These are strong statements that cannot go without citations to support them. Secondly, the authors should stick to the standard name, 'Xpert MTB/RIF' and not GeneXpert MTB/RIF'.
Methods: Section 2.1 - An outlawed word, 'suspected' has been used. This word is prohibited for use in an TB-related publications or communications as it stigmatises victims of TB. It indirectly apportions blame to TB patients for being responsible for contracting TB infection.
Section 2.2 - What was the justification for collecting BAL? The protocol is invasive and needs to be justified. Which route was used for collecting the BAL? mouth or nose? What was the reason for excluding Sputum mixed with saliva, pus, blood, and specimen less than 3-5mL? How were the excessive oropharyngeal contaminants determined?
Section 2.6 – Data analysis methodology narrative is so summarised, making it difficult to understand what statistical tests were done and on what comparisons.
Results/discussion/conclusion: Exact p values should be presented. LPA is presented as more sensitive and specific than Xpert MTB/RIF for drug resistant TB diagnosis, but the discussion and conclusion are only focused and in favour of Xpert MTB/RIF. Can the authors explain where LPA results is not discussed and why Xpert MTB/RIF is preferred over LPA?
Round 2
Reviewer 1 Report
the articicle has certainly improved; some of my earlier comments are not well accomodated or considered. you mau still give it a chance
1 Introduction
· The statement: “It is well known that AFB smear-negative TB patients are the major source of spreading TB to healthy individuals when left untreated” is definitely not true. The infectiousness of AFB smear positive patients is much higher . REFERENCE 8 STATES THAT Almost 13% of TB transmission occurs with smear-negative, culture-positive TB patients; 13% IS NOT THE MAJOR SOURCE , THE MAJOR SOURCE SPREADING TB ARE THE SMEAR POSITIVE PATIENTS
· Also the statement: “The continuously increasing frequency of TB in developing countries like Nepal…”is not true. According to WHO global report 2021 the estimated burden of TB in Nepal is decreasing since 2000 IS DELETED?
2.1 How many patient in total were attending the clinic ? NOT ANSWERED, i mean total atteded; total smear positive'total smear negative
3. Results:
Out of 1895 samples collected, 222 smear-negative samples (from 222 patients?) were taken as the study population. This is 12% Were all remaining 1673 patients smear positive? Not clear how these 222 patients were selected: where these the only samples smear negative?, pls make clear STILL NOT CLEAR
Also helpful would be to mention the definition of “smoking” and “alcohol use” and in the methods how it was ascertained STILL TO BE DONE
4 Discussion:
The study shows that “In the current study, Xpert MTB/RIF assay showed low sensitivity (73%)”..also the authors note that smoking and alcohol use are risk factors: the study may improve if sensitivity is shown by different risk groups including age and gender. THIS IS NOT DONE (THE COMMENTS IS DELETED) BUT NUMBERS BECOME SMALL TO SHOW SIGNIFICANCY; YOU MAY MENTION
5. Conclusion
The statement “Several smear negative samples when examined by Xpert MTB/RIF, LPA and culture methods exhibited AFB positive results …..” is not true ; smear negative samples are by definition AFB negative and not AFB positive, I guess the authors mean that several smear negative samples, are sill culture positive ; in other words, patients of whom the samples are AFB negative could still have active TB.
Try to link the conclusion with the main problems mentioned in the introduction: there are many (M)DR missed but when applying Xpert to all symptomatic patients attending the clinic, this proportion may reduce to …..% you may still try
